# Targeted Large-Volume Lymphocyte Removal Using Magnetic Nanoparticles in Blood Samples of Patients with Chronic Lymphocytic Leukemia: A Proof-of-Concept Study

**DOI:** 10.3390/ijms24087523

**Published:** 2023-04-19

**Authors:** Stefanie Janker, Simon Doswald, Roman R. Schimmer, Urs Schanz, Wendelin J. Stark, Martin Schläpfer, Beatrice Beck-Schimmer

**Affiliations:** 1Institute of Anesthesiology, University Hospital Zurich, University of Zurich, 8091 Zurich, Switzerland; 2Institute of Physiology, University of Zurich, 8057 Zurich, Switzerland; 3Institute for Chemical and Bioengineering, ETH, 8093 Zurich, Switzerland; 4Department of Medical Oncology and Hematology, University Hospital Zurich, University of Zurich, 8091 Zurich, Switzerland

**Keywords:** specific cell removal, nanoparticles, tumor cell elimination, blood purification

## Abstract

In the past, our research group was able to successfully remove circulating tumor cells with magnetic nanoparticles. While these cancer cells are typically present in low numbers, we hypothesized that magnetic nanoparticles, besides catching single cells, are also capable of eliminating a large number of tumor cells from the blood ex vivo. This approach was tested in a small pilot study in blood samples of patients suffering from chronic lymphocytic leukemia (CLL), a mature B-cell neoplasm. Cluster of differentiation (CD) 52 is a ubiquitously expressed surface antigen on mature lymphocytes. Alemtuzumab (MabCampath^®^) is a humanized, IgG1κ, monoclonal antibody directed against CD52, which was formerly clinically approved for treating chronic lymphocytic leukemia (CLL) and therefore regarded as an ideal candidate for further tests to develop new treatment options. Alemtuzumab was bound onto carbon-coated cobalt nanoparticles. The particles were added to blood samples of CLL patients and finally removed, ideally with bound B lymphocytes, using a magnetic column. Flow cytometry quantified lymphocyte counts before, after the first, and after the second flow across the column. A mixed effects analysis was performed to evaluate removal efficiency. *p* < 0.05 was defined as significant. In the first patient cohort (*n* = 10), using a fixed nanoparticle concentration, CD19-positive B lymphocytes were reduced by 38% and by 53% after the first and the second purification steps (*p* = 0.002 and *p* = 0.005), respectively. In a second patient cohort (*n* = 11), the nanoparticle concentration was increased, and CD19-positive B lymphocytes were reduced by 44% (*p* < 0.001) with no further removal after the second purification step. In patients with a high lymphocyte count (>20 G/L), an improved efficiency of approximately 20% was observed using higher nanoparticle concentrations. A 40 to 50% reduction of B lymphocyte count using alemtuzumab-coupled carbon-coated cobalt nanoparticles is feasible, also in patients with a high lymphocyte count. A second purification step did not further increase removal. This proof-of-concept study demonstrates that such particles allow for the targeted extraction of larger amounts of cellular blood components and might offer new treatment options in the far future.

## 1. Introduction

Chronic lymphocytic leukemia (CLL) is a mature B-cell neoplasm with a monoclonal lymphocytic leukocytosis ≥5 G/L [1]. The incidence of the most common type of leukemia increases with age [2]. The disease is typically diagnosed in advanced stages due to its indolent nature. Treatment options include watch-and-wait, standard chemo-, immune-, or chemoimmunotherapy, as well as tyrosine kinase inhibitors, whereby only allogeneic stem cell transplantation is potentially curative [2]. Depending on patient frailty and fitness, the treatment goals may vary from a curative approach to good remission with limited toxicity or palliation [3,4]. The latter is typically the case in individuals with older age, complicating comorbidities or organ dysfunction to avoid side effects of systemic therapies. In such patients, disease progression is associated with increasing lymphocytosis, lymphadenopathy, spleno- and hepatomegaly and peripheral blood cytopenia due to bone marrow displacement by the neoplastic B-cell infiltration.

To date, leukapheresis is the only form of peripheral bulk cytoreductive treatment option in lymphoid neoplasms. This is an extracorporeal cell separation procedure based on each cell type’s specific density. Therefore, we aimed to develop a novel, potential treatment option by removing cluster of differentiation (CD) 52-expressing B lymphocytes from blood samples of CLL patients using a formerly clinically proven agent. Most important, the project has proof of principle character focusing on the feasibility of extracting higher amounts of tumor cells with magnetic nanoparticles. So far, rare circulating tumor cells have been successfully eliminated using a magnetic nanoparticles [5].

Alemtuzumab (MabCampath^®^) is a humanized monoclonal IgG1κ antibody targeting CD52. This cell surface marker is highly expressed in B- and T cells. MabCampath^®^ was a formerly approved agent (which is presently still available for this indication by an Access Program of Clinigen) for the intravenous treatment of CLL in the relapsed and refractory setting [6]. It is, therefore, still an ideal candidate for further clinical evaluation within our ex vivo setting. Compared to leukapheresis, using alemtuzumab-coated magnetic nanoparticles is a targeted cellular removal, which decreases treatment time and could reduce systemic side effects. As nanoparticles can be personalized, this approach may be adapted in other states of clinically significant hyperleukocytosis (leukostasis) in which leukapheresis is indicated [7].

Current medical applications of nanoparticles include the targeted transport of specific molecules, even crossing natural barriers, such as the blood-brain barrier [8]. Our group has previously developed particles capable of removing small inorganic substances from whole blood, such as cytokines (interleukin-6, IL-6) [9], lipopolysaccharides [10,11,12], digoxin and lead [13] or single tumor cells [5].

In hyperleukocytosis, large numbers of cells must be removed. Here, we engineered magnetic, carbon-coated nanoparticles linked to alemtuzumab intending to reduce B lymphocyte counts in CLL patients ex vivo and, therefore, potentially offer a novel approach to disease control in patients in whom curative treatment is not feasible.

## 2. Results

Table 1 reports the patient characteristics of the first cohort working with a fixed amount of nanoparticles. Table 2 shows the patient characteristics of the second cohort when the number of nanoparticles was doubled in patients with a lymphocyte count > 20 G lymphocytes/L blood.

### 2.1. CD19-Positive B Lymphocyte Removal Using Anti-CD52-Coated Nanoparticles

In the first cohort, using a fixed nanoparticle concentration (*n* = 10), B lymphocytes were reduced by 38% after one purification step (*p* = 0.002 compared to PBS). The second purification allowed a further reduction of 15% to 47% (*p* = 0.005 compared to the lymphocyte number after the first purification step), Figure 1A.

In patients with ≤20 G lymphocytes/L blood, the first purification reduced B lymphocytes to 44% (*p* = 0.002 compared to PBS) and the second purification to 16%, however without statistical significance (*p* = 0.053 compared to the lymphocyte number after the first purification step with *n* = 5) of the PBS control, Figure 1B.

B lymphocyte removal (relative reduction) was less efficient in patients with a high lymphocyte count. In patients with >20 G lymphocytes/L blood, B lymphocytes were reduced to 81% (*p* = 0.027 compared to PBS) and 71% (*p* = 0.111 compared to the lymphocyte number after the first purification step) of the PBS control, Figure 1C. Detailed data regarding cleaning steps referring to each patient is provided in the Appendix A.

### 2.2. CD19-Positive B Lymphocyte Removal Using Anti-CD52-Coated Nanoparticles Adapted to Lymphocyte Count

For another *n* = 11 cases (second cohort), the number of nanoparticles was doubled in patients with >20 G lymphocytes/L blood to determine whether increased nanoparticle concentration would result in more efficient lymphocyte removal. The first purification step removed 44% (*p* < 0.001 compared to PBS) of CD19-positive B lymphocytes of the PBS control. Adding a second purification step did not increase removal efficacy (*p* = 0.927 compared to the lymphocyte number after the first purification step; Figure 2A).

If patients with low and high lymphocyte counts were evaluated separately, the following results were obtained: In patients with ≤20 G lymphocytes/L blood (*n* = 5), B lymphocytes were reduced to 54% (*p* = 0.031 compared to PBS) and remained at 55% (of the PBS control) when a second purification step was applied (*p* = 0.964 compared to the lymphocyte number after the first purification step), Figure 2B.

In patients with >20 G lymphocytes/L blood (*n* = 6), B lymphocytes were reduced to 62% (*p* = 0.003 compared to PBS). They remained at 60% (*p* = 0.760 compared to the lymphocyte number after the first purification step) after the first and second purification steps, respectively, Figure 2C. This purification procedure with a higher nanoparticle concentration was more efficient compared to the use of fewer particles (Figure 1C) (reduction by approximately 40% vs. 20% after the first purification step and 40% vs. 30% after the second step). Detailed data regarding cleaning steps referring to each patient is provided in the Appendix A.

**Figure 2 ijms-24-07523-f002:**
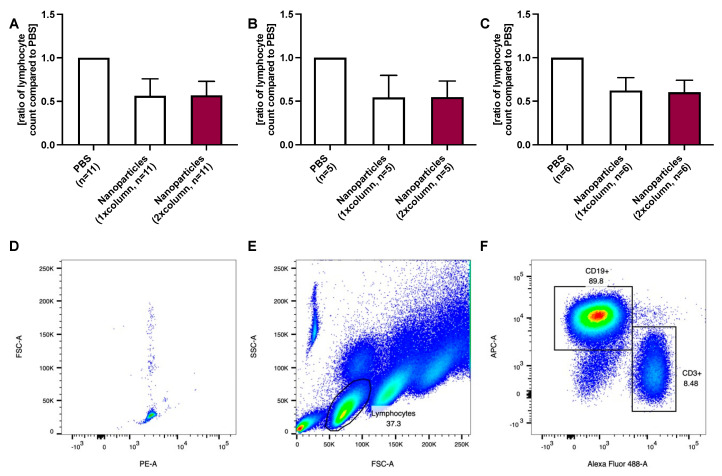
(**A**) shows the relative concentration of CD19-positive B lymphocytes to unpurified control after 1 and 2 alemtuzumab-nanoparticle purification treatments over a magnetic column. (**B**) The removal efficiency in patients with a lower lymphocyte count (≤20 G/L blood) was similar to the purification in patients with a (**C**) higher lymphocyte count despite an increase in nanoparticle concentration in these patients. An example of gating is given in D-F from a patient with more than 20 G lymphocytes/L. 5000 events were recorded using counting beads, gated with FSC-A and PE-A) (**D**). Gating of the lymphocyte population was performed using the side scatter (SSC-A) and FSC-A (**E**) as described in the literature [14], followed by the exclusion of doubles using FSC-A and FSC-H. CD19+ B- and CD3+ T-lymphocyte populations were then gated using APC-A (bound to the CD19-antibody) and Alexa fluor 488 (bound to the CD3-antibody) (**F**).

### 2.3. CD52-Positive Lymphocyte Removal Using Anti-CD52-Coated Nanoparticles Adapted to Lymphocyte Count

In a second measurement of cohort 2, alemtuzumab-coated magnetic nanoparticles were tested regarding their targeted CD52-positive lymphocyte removal efficiency (all cells expressing CD52) (*n* = 11 patients). In individuals with >20 G lymphocytes/L blood, nanoparticle concentration was doubled, as described above.

On average, CD52-positive cells were reduced to 66% (*p* < 0.001 compared to PBS) after the first and to 60% (*p* = 0.041 compared to the lymphocyte number after the first purification step) after the second purification of the unpurified PBS control (Figure 3A).

In individuals with ≤20 G lymphocytes/L blood, 62% and 57% of CD52-positive lymphocytes (*p* = 0.017 compared to PBS and *p* = 0.314 compared to the lymphocyte number after the first purification step) of the PBS control remained in the blood (Figure 3B).

In the group with >20 G lymphocytes/L blood, 72% (*p* = 0.009 in comparison to PBS) and 54% (*p* = 0.224 compared to the lymphocyte number after the first purification step) of CD52-positive cells remained in the sample (compared to PBS control; Figure 3C).

CD19-positive cell removal was comparable to CD52-positive cell elimination. Detailed information regarding the cleaning steps of each patient is provided in the Appendix A.

**Figure 3 ijms-24-07523-f003:**
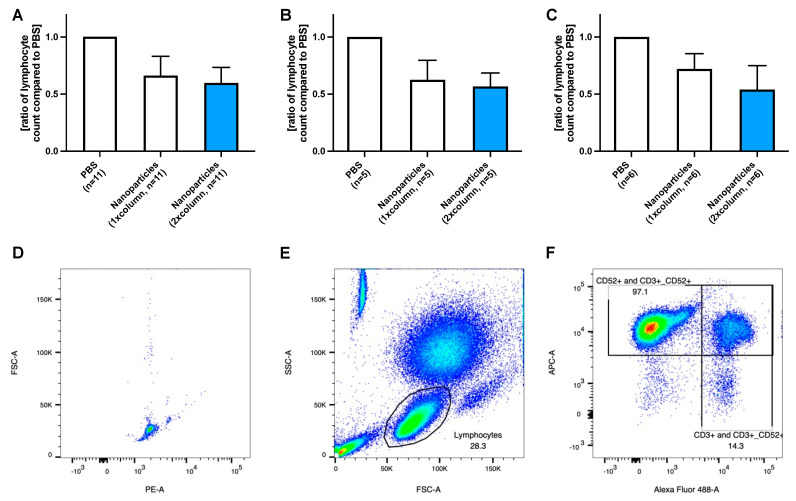
(**A**) shows the average relative concentration of CD52 positive lymphocytes to unpurified control after 1 and 2 alemtuzumab-nanoparticle purification treatments. (**B**) The removal efficiency in patients with a lower lymphocyte count (≤20 G/L blood) was similar to the purification in patients with a (**C**) higher lymphocyte count despite doubling the nanoparticles in these patients. For the analysis, 5000 events were recorded using counting beads, gated with FSC-A and PE-A (**D**). Gating of the lymphocyte population was performed using the side scatter (SSC-A) and FSC-A (**E**) as described in the literature [14], followed by the exclusion of doubles using FSC-A and FSC-H. While a CD52 (marked with APC) and a CD3 (marked with Alexa Fluor 488) staining was performed, this experiment focused on the evaluation of CD52-positive cells irrespective of the CD3 staining (**F**).

## 3. Discussion

This is a proof-of-concept study of targeted B lymphocyte removal using alemtuzumab-coupled nanoparticles in blood samples of CLL patients ex vivo. On average, a single purification step using these nanoparticles resulted in a 50% lymphocyte reduction, while a second purification step did not significantly increase removal. However, adding a higher nanoparticle concentration to blood samples from patients with high lymphocyte count (>20 G lymphocytes/L blood) improved the efficiency of lymphocyte removal. This study had a main focus on B lymphocytes as nanoparticles were tested in blood from patients suffering from a B cell neoplasm. To evaluate a second large population of blood cells expressing CD52, purification of CD3-positive T lymphocytes was assessed, which showed similar results as for B lymphocyte removal (data are provided in the Appendix A).

To date, lymphocyte removal using a magnetic nanoparticle-based approach has not been demonstrated before. Therefore, a comparison to similar projects is not possible. We published a proof-of-principle study showing magnetic nanoparticle purification of circulating tumor cells [5]. While only sparse circulating tumor cells per blood volume are found, the scenario is quite different in patients with CLL, with a vast majority of tumor cells concerning the composition of blood cells. Our findings highlight the feasibility of removing tumor cells from the blood spanning from small to large quantities of cells. In the study, unspecific interactions of this nanoparticle type, covalently bound to an antibody, could be largely excluded [5]. Of utmost importance is the targeted antigen-based approach through either the surface cell marker EpCAM (epithelial cell adhesion molecule, CD326) or CD52.

Future studies must evaluate lymphocyte removal in CCL patients using an extracorporeal device and determine the effectiveness of this method. In theory, antibody-directed lymphocyte reduction may be superior to standard leukapheresis in various states of hyperleukocytosis (not resulting in leukostasis in CLL) as the nanoparticles may be modified according to the patient’s leukocyte surface characteristics allowing for a targeted cell removal as opposed to an unspecific leukocyte depletion [15,16]. Furthermore, the generalizability and transferability of this method to target and remove specific cells in the blood, especially in other types of leukemia, where hyperleukocytosis often results in symptomatic leukostasis, is undoubtedly a strength of this method and opens new medical treatment approaches.

The relative reduction of lymphocytes in patients with high lymphocyte counts was less pronounced than in patients with a lower lymphocyte count when using a fixed nanoparticle concentration. However, purification efficiency was increased after optimizing the nanoparticle-to-target ratio in patients with high lymphocyte counts, adding more nanoparticles to the blood samples. However, the results are still not as promising as in the presence of lower lymphocyte counts. So far, the reasons for this phenomenon remain elusive. A possible explanation might be the increased viscosity of the blood in CLL patients. Previous studies have claimed that in contrast to acute leukemias, the viscosity of the blood seems to be only altered at lymphocyte counts > 100 G/L in chronic lymphocytic leukemia [17]. However, many pro-inflammatory genes in CLL are known to be upregulated [18], and chronic inflammation increases blood viscosity [19]. Moreover, it has been described that the deformability of leukocytes in CLL is reduced compared to leukocytes in healthy individuals [20]. Even a minimal increase in viscosity combined with reduced deformability of leukocytes might inhibit their extraction rate by alemtuzumab-coupled nanoparticles. Additionally, the number of surface antigens (CD52 in this study) and the antibody binding force may impact removal efficiency. To improve removal efficiency, it might be necessary to remove not only cells based on a single but on a second surface antigen, dependent on the cell characteristics of target cells.

As with any experimental approach, this study has strengths and limitations. The first limitation is that only a small cohort of patients was analyzed. A second limitation refers to a missing sample size calculation. This is justified as no preliminary data in the literature are available. Any further interpretation should be avoided as the sample size does not allow it.

A strength of this study is this new innovative approach which lays the ground for further research in a larger cohort. Additionally, using a clinically established drug, such as alemtuzumab, brings the advantage of dealing with a well-known substance. However, binding a compound to nanoparticles might be related to new side effects, which should be carefully evaluated. The most crucial strength of using magnetic nanoparticles is the fact that the purification process would be an extracorporeal approach. This way, once nanoparticles are magnetically removed, they would not find access to the body and, therefore, would not be a risk for the patient. Additionally, side effects after the intravenous application of alemtuzumab, such as fever, nausea, vomiting, dyspnea or hypotension, would not be found with an extracorporeal approach [21].

## 4. Material and Methods

### 4.1. Patients

This ex vivo trial was approved by the competent ethics committee (Cantonal ethics committee, Zurich, Switzerland, approval number 2016-01140; date: 21 November 2016, signed by Erich W. Russi and Peter Kleist), and the study was registered on clinicaltrials.gov NCT04290923. All interventions were performed according to the declaration of Helsinki and according to legal requirements. Patients, at least 18 years old, were eligible if suffering from CLL without prior treatment (chemo- and/or immunotherapy or hematopoietic cell transplantation) defined as mature B-cell neoplasm with monoclonal lymphocytic leukocytosis ≥ 5 G/L. Written and informed consent was obtained before study inclusion. Patients were excluded if they presented with any of the following: ethical issues, inability to follow procedures of the research project (i.e., due to language barrier, psychological disorders, or dementia). This trial is reported according to the revised standards for quality improvement reporting excellence guidelines (SQUIRE 2.0) [22]. For two experimental approaches, the following groups were defined: cohort 1, consisting of 10 patients and cohort 2, with 11 patients.

Blood was collected in heparin tubes (BD Vacutainer^®^, Becton Dickinson GmbH, Heidelberg, Germany) during a routine visit to the Department of Medical Oncology and Hematology at the University Hospital Zurich.

### 4.2. Nanoparticles

Carbon-coated, alemtuzumab-bound nanoparticles were produced in a multi-step process by the Department of Chemistry at ETH Zurich, Switzerland. The functionalization and manufacturing process of the particles has been described in detail in previous projects [23,24], with the difference that alemtuzumab antibodies were covalently bound to the carbon groups: CCo@PG-COO-alemtuzumab conjugates were synthesized via the intermediates Cco@phenylethanol, Cco@polyglycdiol (PG), and Cco@PG-COOH. The detailed synthesis is described in the Appendix A.

### 4.3. Ex Vivo Lymphocyte Removal

#### 4.3.1. Experimental Approach

Before adding alemtuzumab-coupled nanoparticles to blood samples, they were sonicated in an ice-cooled water bath. For every 250 µL of patient blood, 25 µL of magnetic nanoparticles (5 mg/mL) or 25 µL of phosphate-buffered saline (PBS 1x, pH 7.2, Gibco^®^, Life Technologies^TM^, Hopkinton, MA, USA) were added. PBS was chosen as the negative control because a previous study has revealed that not functionalized nanoparticles have unspecific absorption properties and that IgG bound to functionalized nanoparticles does not impact nanoparticle-based magnetic cell removal [5], which could be confirmed in the blood of healthy volunteers within the current setting with IgG isotype- and anti-CD52-coated nanoparticles (Appendix A). The nanoparticle concentration was determined in a master thesis preceding the current project. Blood samples were placed on a rocker for 2 min and were then pipetted onto a magnetic column (MACS^®^ cell separator columns, MACS^®^ Miltenyi Biotec, Bergisch Gladbach, Germany), each pre-equilibrated with 500 µL PBS. Two washing steps were performed, the first with 750 µL and the second with 500 µL PBS. The flow-through was collected, followed by a second removal step. Results show lymphocyte counts before, after the first, and after the second removal.

#### 4.3.2. Variation with Nanoparticle Concentration

In the first cohort, 25 µL of magnetic nanoparticles (5 mg/mL) were added to the blood samples independent of the lymphocyte count (results presented in Figure 1), while in the second cohort, 50 µL of magnetic nanoparticles were added if the lymphocyte count was >20 G lymphocytes/L (results presented in Figure 2 and Figure 3). The lymphocyte count was determined at the hematology laboratory of the University Hospital Zurich, Switzerland.

### 4.4. Staining of Lymphocytes and Cell Counting

T lymphocytes were labeled with a fluorescent CD3, B lymphocytes with a CD19, and CD52 positive lymphocytes with a CD52 antibody. Samples were processed and analyzed by flow cytometry (FACS) in research laboratories at the University Hospital Zurich, Switzerland.

Detailed antibody information is provided in the Appendix A. In brief, 1 ug of each primary antibody was added. The samples were incubated at 4 °C for 30 min. Then, 10 mL red blood cells, RBC) Lysis Buffer (eBioscience™ 1X RBC Lysis Buffer, Invitrogen™, Thermo Fisher Scientific, Waltham, MA, USA), diluted 1:10 with ultrapure water, was added and allowed to incubate in the dark at room temperature for 10 min. After centrifugation (400 g, room temperature, 5 min), the supernatant was removed, and the cell pellet was resuspended in 200 µL PBS and fixed in 200 µL formalin (4%, buffered). Before FACS analysis, 25 µL flow cytometry cell counting beads (0.54 × 10^5^ beads/50 µL, Thermo Fisher Scientific, Waltham, MA, USA) were added.

FACS samples were measured on BD FACS Canto^TM^ II (Becton&Dickinson (BD), Franklin Lakes, NJ, USA). The data evaluation was performed using FlowJo Version 10.8.1 (BD, Franklin Lakes, NJ, USA). First, 5000 events were recorded using counting beads. Then, the lymphocyte population was gated using the forward- and the side-scatter as described in the literature [14]. In the next step, CD3, CD19 and CD52 positive cells were gated and quantified using the corresponding fluorochrome on the individual antibodies.

In the first two experiments, the focus was on B lymphocytes, while in the third one, all CD52-expressing lymphocytes were targeted.

### 4.5. Statistical Analyses

Data were evaluated in GraphPad Prism 8 (GraphPad Software, La Jolla, CA, USA). Values are presented as mean ± standard deviation (SD). The groups were compared using a mixed-effect model and Holm-Sidak’s multiple comparison test. A *p*-value < 0.05 was determined to be statistically significant.

This section may be divided into subheadings. It should provide a concise and precise description of the experimental results, their interpretation, as well as the experimental conclusions that can be drawn.

## 5. Conclusions

In conclusion, our data demonstrate that a significant lymphocyte count reduction is feasible in CLL patients using alemtuzumab-coupled nanoparticles. The study shows the successful and targeted magnetic nanoparticle-based removal of cellular blood components present in a large volume. In contrast to previous experiments in which relatively sparse circulating tumor cells were removed [5], this proof of concept study was able to relevantly eliminate a high number of lymphocytes in blood samples of CLL patients. Future studies will have to be performed to validate this approach. The setting can be further developed to achieve extracorporeal blood purification. This approach might be a promising palliative treatment for CLL patients. Furthermore, extrapolating this approach, changing particles to allow targeted extraction of any blood component potentially offers novel revolutionary treatment options in other types of leukemia.

## Figures and Tables

**Figure 1 ijms-24-07523-f001:**
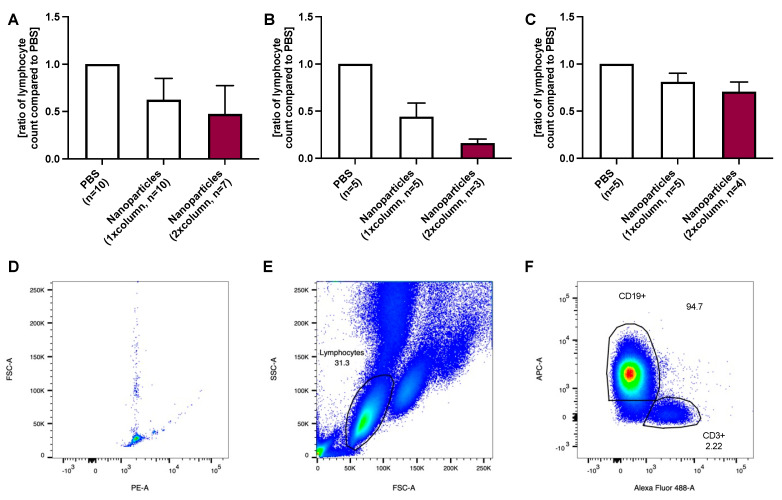
(**A**) shows the effect of alemtuzumab-nanoparticle treatment on CD19-positive B lymphocytes after 1 and 2 purification steps over a magnetic column. (**B**) In patients with a lower lymphocyte count (≤20 G/L blood), the purification appears to be more efficient than in (**C**) patients with a higher lymphocyte count. An example of gating is given in (**D**–**F**). Five thousand events were recorded using counting beads. The forward scatter (FSC-A) and phycoerythrin (PE-A) were plotted (**D**). Gating of the lymphocyte population was performed using the side scatter (SSC-A) and FSC-A (**E**) as described in the literature [14], followed by the exclusion of doubles by using forward scatter area (FSC-A) and forward scatter height (FSC-H). CD19+ B- and CD3+ T-lymphocyte populations were then gated using APC-A (bound to the CD19-antibody) and Alexa fluor 488 (bound to the CD3-antibody) (**F**).

**Table 1 ijms-24-07523-t001:** Patient characteristics of cohort 1.

	≤20 G Lymphocytes/L Blood (*n* = 4)	>20 G Lymphocytes/L Blood (*n* = 6)
Age [years] (mean ± SD)	71 ± 8.79	61 ± 10.39
Sex Female (*n*) Male (*n*)	3 1	2 4
Chronic lymphocytic leukemia (*n*)	4	6
Patients under watch-and-wait strategy (*n*)	3	5
Patients under first-line treatment (*n*)	1	0
Patients under second-line treatment (*n*)	0	1
Lymphocyte count [G/L] (mean ± SD)	9.14 ± 5.10	65.57 ± 26.14

Legend: *n*: numbers, SD: standard deviation.

**Table 2 ijms-24-07523-t002:** Patient characteristics of cohort 2.

	≤20 G Lymphocytes/L Blood (*n* = 4)	>20 G Lymphocytes/L Blood (*n* = 6)
Age [years] (mean ± SD)	76 ± 7.44	62 ± 8.23
Sex Female (*n*) Male (*n*)	2 3	4 2
Chronic lymphocytic leukemia (*n*)	5	6
Patients under watch-and-wait strategy (*n*)	4	5
Patients under first-line treatment (*n*)	1	0
Patients under second-line treatment (*n*)	0	1
Lymphocyte count [G/L] (mean ± SD)	11.87 ± 4.22	71.12 ± 20.25

Legend: *n*: numbers, SD: standard deviation.

## Data Availability

Anonymized study data are available from the corresponding author upon reasonable request.

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
