# Peer review of "Targeted Large-Volume Lymphocyte Removal Using Magnetic Nanoparticles in Blood Samples of Patients with Chronic Lymphocytic Leukemia: A Proof-of-Concept Study"

_ijms, 2023, doi:10.3390/ijms24087523_

Round 1

Reviewer 1 Report

The article entitled “Targeted large-volume lymphocyte removal using magnetic nanoparticles in blood samples of patients with chronic lymphocytic leukemia” by Stefanie Janker et al. reported a new technological approach for removing excessive number of lymphocytes in CLL. This technique compared to leukapheresis is more specific and opens possibilities to use the technology for removing malignant cells in other diseases. Similar techniques have been tested in the past using lectins instead of monoclonal antibodies to remove cells from the blood but did not reach this step of development. It may be of interest to refer to the techniques used for leukodepletion in blood transfusion.

This article is well structured, easy to read and the conclusions are well balanced. It is obviously a first study which included a limited number of patients but very encouraging to pursue this attempt.

Minor revisions:

The figures may be improved.

-          First the rel. to PBS may be expressed as a ratio of lymphocyte count compared to PBS.

-          The histogram which is the first run may have a different color and no blank.

-          The flow cytometry images are colored but it would be logical to separate filtrations techniques and flow analysis or presented as supplementary results showing different experiments and not just a typical result.

Author Response

Reviewer 1

Comments and Suggestions for Authors

The article entitled “Targeted large-volume lymphocyte removal using magnetic nanoparticles in blood samples of patients with chronic lymphocytic leukemia” by Stefanie Janker et al. reported a new technological approach for removing excessive number of lymphocytes in CLL. This technique compared to leukapheresis is more specific and opens possibilities to use the technology for removing malignant cells in other diseases. Similar techniques have been tested in the past using lectins instead of monoclonal antibodies to remove cells from the blood but did not reach this step of development. It may be of interest to refer to the techniques used for leukodepletion in blood transfusion.

This article is well structured, easy to read and the conclusions are well balanced. It is obviously a first study which included a limited number of patients but very encouraging to pursue this attempt.

The authors thank the reviewer for their time in evaluating the manuscript and for providing feedback.

Minor revisions:

The figures may be improved.

-          First the rel. to PBS may be expressed as a ratio of lymphocyte count compared to PBS.

Changed according to the reviewer’s suggestion.

-          The histogram which is the first run may have a different color and no blank.

If the authors understand the reviewer correctly, the reviewer would appreciate being able to see the change after the first and second purification steps. The figures just demonstrate the measurement technique. For transparency reasons, the full measurement data (first and second flow cytometry data) are now provided for all experiments in the supplementary material.

-          The flow cytometry images are colored but it would be logical to separate filtrations techniques and flow analysis or presented as supplementary results showing different experiments and not just a typical result.

Magnetic elimination was measured by flow cytometry. Therefore, the authors consider it to be important to explain, with an example, how they have measured the data. The authors understand, however, the problem of showing a single experiment as an example. Therefore, the authors have now uploaded all flow cytometry measurements in supplementary documents.

Reviewer 2 Report

The article is a nice example of proof-of-principle trial. Authors reported the results of the experimental study on novel targeted magnetic lymphocyte removal technic based on CD52 antibodies bounded on nanoparticles. Besides the promising results and sufficient scientific soundness, several improvements are required before considering the article for publication:

  1. The authors didn’t state clear what was used as a control group of samples for nanoparticles treatment. The paragraph 2.3 contains the information that either 25 μl of magnetic nanoparticles or 25 μl of phosphate-buffered saline was added to each blood samples. Thus, the authors should specify the reason for choosing PBS instead of more sophisticated control (for ex. Uncoated nanoparticles).
  2. The graphs (D,E,F) and legend on Figure 1 should be ‘self-readable’: the authors are recommended to concretize in the legend what exactly was labeled with  Alexa fluor 488.
  3. Did the authors check the effect of nanoparticles on erythrocytes?  Was an any evidence of their lysis detected?

Author Response

Reviewer 2

The authors thank the reviewer for their time in evaluating the manuscript and for their constructive feedback.

The article is a nice example of proof-of-principle trial. Authors reported the results of the experimental study on novel targeted magnetic lymphocyte removal technic based on CD52 antibodies bounded on nanoparticles. Besides the promising results and sufficient scientific soundness, several improvements are required before considering the article for publication:

  1. The authors didn’t state clear what was used as a control group of samples for nanoparticles treatment. The paragraph 2.3 contains the information that either 25 μl of magnetic nanoparticles or 25 μl of phosphate-buffered saline was added to each blood samples. Thus, the authors should specify the reason for choosing PBS instead of more sophisticated control (for ex. Uncoated nanoparticles).

The authors agree with the reviewer: The choice of negative control was inadequately explained. Uncoated nanoparticles have different anti-fouling properties and may, due to unspecific protein absorptions, be a poor negative control (1). Functionalized antibodies to which IgG is bound instead of an antibody against a surface antigen have no effect on cell number (1). For these two reasons as well as for cost arguments, PBS was chosen as a control. This information is now included in the methods section of the manuscript.

  1. The graphs (D,E,F) and legend on Figure 1 should be ‘self-readable’: the authors are recommended to concretize in the legend what exactly was labeled with  Alexa fluor 488.

The missing information is now provided in the corresponding figure legends.

  1. Did the authors check the effect of nanoparticles on erythrocytes?  Was an any evidence of their lysis detected?

Lysis has not been measured (e.g., by free hemoglobin quantification). The authors did not find evidence for erythrocyte lysis, but this statement is based solely on the authors’ observations.

References

  1. Doswald S, Herzog AF, Zeltner M, et al: Removal of Circulating Tumor Cells from Blood Samples of Cancer Patients Using Highly Magnetic Nanoparticles: A Translational Research Project. Pharmaceutics 2022; 14(7)

Reviewer 3 Report

The study from Janker, Doswald and colleagues aims of targeting and removing B lymphocytes in blood samples of chronic lymphocytic leukemia patients, using alemtuzumab-targeted nanoparticles. The manuscript is of interest and well structured. However, and as the authors pointed during the discussion, the small sized cohorts used (n=10, each) make it difficult to reliably corroborate the conclusions from the study. In addition, could the authors argue about:

1.     The percentage of other CD52-targeted lymphocytes besides the CD19-positive B lymphocytes?

2.     The purification procedure with a higher nanoparticle concentration seems to increase the efficiency of cell removal. How did the authors find the optimal concentration of nanoparticles to used/ cell count? Would these concentrations remain optimal for higher number of cells? In terms of numbers, which is the maximum number of cells able to be removed with this approach?

3.  Regarding the optimal concentration of nanoparticles used, the authors stated that “the results are not as promising as in the presence of lower lymphocyte counts”. Could you argue about the strategies to overcome this drawback?

Minor:

Section 4. Materials and Methods appears to be misplaced.

Could the authors increase the resolution of figure 1 from Online supplement 1.

Author Response

Reviewer 3

The authors thank the reviewer for their assessment of this article and for providing feedback.

The study from Janker, Doswald and colleagues aims of targeting and removing B lymphocytes in blood samples of chronic lymphocytic leukemia patients, using alemtuzumab-targeted nanoparticles. The manuscript is of interest and well structured. However, and as the authors pointed during the discussion, the small sized cohorts used (n=10, each) make it difficult to reliably corroborate the conclusions from the study. In addition, could the authors argue about:

  1. The percentage of other CD52-targeted lymphocytes besides the CD19-positive B lymphocytes?

In this work, the main focus lies on CD19-positive B lymphocytes based on our hypothesis referring to blood from patients suffering from a B cell neoplasm. The aim of the study was to show that we are able to remove large amounts of cells contrary to the scenario of single circulating tumor cell elimination. Also, CD3-positive T cell removal was evaluated as these cells represent a major cell population in human blood. However, additional CD52-positive cell populations were not studied. The authors included this statement in the discussion of the manuscript. Data and additional graphs of CD3-positive T lymphocyte removal are provided in the online supplement.

  1. The purification procedure with a higher nanoparticle concentration seems to increase the efficiency of cell removal. How did the authors find the optimal concentration of nanoparticles to used/ cell count? Would these concentrations remain optimal for higher number of cells? In terms of numbers, which is the maximum number of cells able to be removed with this approach?

The nanoparticle concentrations were established in dilution series in a master thesis, as mentioned in the manuscript. The maximum removal potential of the particles has not been defined in this setting. We just wanted to show in a first proof-of-principle approach that through an additional simple purification step efficiency of targeted cell elimination can be increased. It is important to highlight that this proof-of-concept trial aims at demonstrating that not only single cells, such as circulating tumor cells but also abundant cells can be removed with the technique suggested. The authors do not feel qualified to hypothesize on the maximum number of cells that can be removed using this approach.

A general remark: The removal potential is mainly depending on the number of nanoparticles, the number of surface antigens on the target cells, and the binding force between antigen and antibody, which is indirectly also defined by the type of liquid and its composition (solid and fluid part). The final application we envisage, a type of nanoparticle dialysis, will be a dynamic purification process with the continuous injection of new nanoparticles into the extracorporeal system to guarantee optimal efficiency.

  1. Regarding the optimal concentration of nanoparticles used, the authors stated that “the results are not as promising as in the presence of lower lymphocyte counts”. Could you argue about the strategies to overcome this drawback?

The authors have now implemented an additional sentence in the discussion that potentially more than one surface antigen should be targeted based on the cell characteristics to be removed.

Minor:

Section 4. Materials and Methods appears to be misplaced.

This has been corrected.

Could the authors increase the resolution of figure 1 from Online supplement 1.

A higher resolution of the electron micrograph has been implemented into the document.

Round 2

Reviewer 3 Report

Regarding the percentage of other CD52-targeted lymphocytes besides the CD19-positive B lymphocytes that could also be removed, the authors replied that “To evaluate a second large population of blood cells expressing CD52 purification of CD3-positive T lymphocytes was assessed, which showed similar results as for B lymphocyte removal (data are provided in the online supplement 3)”. How do the authors argue about the specificity of the strategy?

Author Response

Comments and Suggestions for Authors Regarding the percentage of other CD52-targeted lymphocytes besides the CD19-positive B lymphocytes that could also be removed, the authors replied that “To evaluate a second large population of blood cells expressing CD52 purification of CD3-positive T lymphocytes was assessed, which showed similar results as for B lymphocyte removal (data are provided in the online supplement 3)”. How do the authors argue about the specificity of the strategy?   Thank you for this question about the specificity of the nanoparticles. Please find attached the results. Blood of healthy volunteers was subjected to the following treatment options: adding phosphate-buffered saline (PBS), IgG- or anti-CD52 coupled nanoparticles. Experiments were performed as described. After an incubation time the samples were run over a magnetic column, and the remaining lymphocytes were determined by flow cytometry. The tests revealed that IgG-coupled particles did not remove any cells, as demonstrated in the figure below.  Concerning specificity for B lymphocyte removal: the authors did not claim the procedure was specific for B lymphocyte removal. This was also not the aim of the study. The aim was to demonstrate a procedure that allowed for removing high cell counts (as the target was CD 52, thus a marker for mature lymphocytes and not specifically to B lymphocytes).